# *N*-Acetylcysteine Protects against the Anxiogenic Response to Cisplatin in Rats

**DOI:** 10.3390/biom9120892

**Published:** 2019-12-17

**Authors:** Rade Vukovic, Igor Kumburovic, Jovana Joksimovic Jovic, Nemanja Jovicic, Jelena S. Katanic Stankovic, Vladimir Mihailovic, Milos Djuric, Stefan Velickovic, Aleksandra Arnaut, Dragica Selakovic, Gvozden Rosic

**Affiliations:** 1Clinic for Anesthesiology and Critical Care, Military Medical Academy, Crnotravska 17, 11000 Belgrade, Serbia; radvuk@gmail.com; 2Department of Physiology, Faculty of Medical Sciences, University of Kragujevac, Svetozara Markovica 69, 34000 Kragujevac, Serbia; ikumburovic@gmail.com (I.K.); jovana_joksimovic@yahoo.com (J.J.J.); 3Department of Histology and Embryology, Faculty of Medical Sciences, University of Kragujevac, Svetozara Markovica 69, 34000 Kragujevac, Serbia; nemanjajovicic.kg@gmail.com; 4Department of Science, Institute for Information Technologies Kragujevac, University of Kragujevac, Jovana Cvijica b.b., 34000 Kragujevac, Serbia; jkatanic@kg.ac.rs; 5Department of Chemistry, Faculty of Science, University of Kragujevac, Radoja Domanovica 12, 34000 Kragujevac, Serbia; vladimir.mihailovic@pmf.kg.ac.rs; 6Institute of Pharmacology, Clinical Pharmacology and Toxicology, Faculty of Medicine, University of Belgrade, Dr Subotica 8, 11000 Belgrade, Serbia; dr.milosdj@gmail.com; 7Department of Dentistry, Faculty of Medical Sciences, University of Kragujevac, Svetozara Markovica 69, 34000 Kragujevac, Serbia; velickovicstefan91@gmail.com (S.V.); sandra11_92@yahoo.com (A.A.)

**Keywords:** cisplatin, *N*-acetylcysteine, oxidative stress, apoptosis, hippocampus, rats

## Abstract

Since cisplatin therapy is usually accompanied with numerous toxicities, including neurotoxicity, that involve tissue oxidative damage, the aim of this study was to evaluate the possible protective effect of *N*-acetylcysteine (NAC) on the anxiogenic response to cisplatin (CIS). Thirty-two male Wistar albino rats divided into four groups (control, cisplatin, NAC, and CIS + NAC). All treatments were delivered intraperitoneally. On day one, the control and cisplatin groups received saline while the NAC and CIS + NAC groups were administered with NAC (500 mg/kg). On the fifth day, the control group received saline while the CIS group was treated with cisplatin (7.5 mg/kg), the NAC group again received NAC (500 mg/kg), and the CIS + NAC group was simultaneously treated with cisplatin and NAC (7.5 and 500 mg/kg, respectively). Behavioral testing, performed on the tenth day in the open field (OF) and elevated plus maze (EPM) tests, revealed the anxiogenic effect of cisplatin that was significantly attenuated by NAC. The hippocampal sections evaluation showed increased oxidative stress (increased lipid peroxidation and decline in antioxidant enzymes activity) and proapoptotic action (predominantly by diminished antiapoptotic gene expression) following a single dose of cisplatin. NAC supplementation along with cisplatin administration reversed the prooxidative and proapoptotic effects of cisplatin. In conclusion, the results obtained in this study confirmed that antioxidant supplementation with NAC may attenuate the cisplatin-induced anxiety. The mechanism of anxiolytic effect achieved by NAC may include the decline in oxidative damage that down regulates increased apoptosis and reverses the anxiogenic action of cisplatin.

## 1. Introduction

Cisplatin, cisplatinum, or cis-diamminedichloroplatinum (II) is a well-known chemotherapeutic drug used to treat a number of different types of cancers including sarcomas, cancers of the soft tissue, muscles, bones, and blood vessels [1]. Although synthesized in 1844, cisplatin has been increasingly interesting since it anticancer activity was shown. Namely, in the 1960s it was discovered to possess cytotoxic properties [2], and in the seventies it became the core compound for the systemic treatment of germ cell cancers. Cisplatin was the first FDA-approved platinum compound for cancer treatment in 1978 [3], which has significantly affected the further development of related platinum-based drugs as potential anticancer treatments [4]. Although highly effective as a cytostatic, cisplatin treatment leads to dose-dependent adverse effects as a consequence of DNA damage, increased production of proinflammatory cytokines, mitochondrial dysfunction, apoptosis, and oxidative stress [5]. These adverse effects include neurotoxicity, hepatotoxicity, nephrotoxicity, ototoxicity, myelosuppression, gastrointestinal toxicity, and cardiotoxicity. Common neurological adverse effects are cognitive deficits, disorientation, visual perception, and hearing disorders [6]. 

Cisplatin exerts its cytotoxic properties by reacting with DNA, which eventually leads to irreversible apoptosis [7]. It binds to the N7 reactive center on purine residues and as such forms DNA–DNA interstrand and intrastrand crosslinks. It is believed that intrastrand adducts are responsible for the cytotoxic effects of cisplatin and the inhibition of DNA replication and transcription. DNA adduct formation is followed by DNA damage recognition by a number of proteins that further transmit DNA damage signals to downstream signaling cascades involving p53, MAPK, and p73, which ultimately induce apoptosis. Furthermore, cisplatin induces apoptosis and inhibits the proliferation of stem cells and neurogenesis in the subventricular and subgranular hippocampal zone by increased expression of proapoptotic genes (*Bid*, *Bik*, *Bax*, *Bok*, *Tp53bp2*, and *Card6*), while reducing the expression of one of the major anti-apoptotic genes, *Bcl*-2 [8]. Cisplatin binds to mitochondrial DNA (mtDNA) causing irreversible damage, leading to inhibitory replication and transcription of mtDNA and subsequent mitochondrial dysfunction and cell death. mtDNA damage and subsequent mitochondrial dysfunction result in the production of free reactive oxygen species (ROS), as well as the induction of oxidative stress [9].

Antioxidants gained significant scientific attention for their prominent beneficial effects that can combat the excessive oxidative stress in organisms. They are compounds that stabilize, scavenge, and suppress the generation of oxidants and free radicals [10]. They can be grouped into enzymatic antioxidants, such as superoxide dismutase SOD, glutathione peroxidase (GSH-Px), and catalase (CAT), and nonenzymatic antioxidants, such as ascorbic acid (vitamin C), α-tocopherol (vitamin E), glutathione (GSH), β-carotene, and vitamin A. Besides several endogenous substances, there are a number of exogenous substances with strong antioxidant properties, such as flavonoids, polyphenols, coenzyme Q10, amino acids, and herbal extracts. They all may exhibit a strong antioxidant and anti-inflammatory potential, however, they differ significantly in their mechanisms of action [5]. *N*-acetylcysteine (NAC) is a sulfur containing amino acid that has recently been proven as a potent antioxidant. It is a more stable form of the L-cysteine amino acid, which is a source of cysteine that is necessary for the synthesis of glutathione, a significant nonenzymatic intracellular antioxidant [11]. As such, NAC may provide possible beneficial results with cisplatin administration in order to prevent cisplatin adverse effects. In addition to its antioxidant properties, NAC may reduce the reactivity of cisplatin within the cell itself. This could be achieved by its direct binding to cisplatin, diminishing its toxicity. Furthermore, it has been recently shown that NAC mitigates cisplatin-induced inhibition of dendritic branching, the formation of oxidative stress, and apoptosis of the cells in the hippocampus, as well as subsequent cognitive dysfunction, without affecting the effectiveness of cytostatic therapy [12].

The objective of this study was to estimate the possible beneficial effects of NAC supplementation along with cisplatin administration in order to prevent the behavioral adverse effects of cisplatin. Furthermore, the study aimed to evaluate the impact of NAC on cisplatin-induced alterations in oxidative stress markers, as well as on the expression of apoptotic genes in the hippocampi of rats. 

## 2. Material and Methods

### 2.1. Animals and Treatment 

Three months old male Wistar albino rats (250–300 g, *n* = 32) were purchased from the Military Medical Academy, Serbia. All animals were kept in transparent cages (four animals per cage) under standard conditions (temperature 23 ± 1 °C, humidity 50 ± 5%) with a light/dark cycle (12/12h) and had free access to food and water. 

The rats were randomly assigned into four equal groups: control, cisplatin (CIS), NAC, and CIS + NAC groups (*n* = 8). As shown in Table 1, at the start of the trial (day 1) the control and cisplatin groups received saline (approximately 2 mL i.p.), while the NAC and CIS + NAC groups were administered with NAC (500 mg/kg i.p., Sigma-Aldrich, Munich, Germany). On the fifth day, the control group received saline (in the same manner), the CIS group was treated with cisplatin (7.5 mg/kg i.p.) (Merck, Paris, France), the NAC group was administered with NAC again (500 mg/kg i.p.), and the CIS + NAC group was simultaneously treated with cisplatin and NAC (7.5 and 500 mg/kg i.p., respectively). 

All research procedures were carried out in accordance with the European Directive for the welfare of laboratory animals No 86/609/EEC, the principles of Good Laboratory Practice (GLP), and in accordance with the ARRIVE guidelines. All experiments were approved by the Ethical Committee of the Faculty of Medical Sciences, University of Kragujevac, Serbia.

### 2.2. Behavioral Testing

Behavioral tests were conducted on the tenth day. At approximately 0900 the rats were placed in the testing room and acclimatized for 1 h before testing in the open field (OF) and the elevated plus maze (EPM) test, respectively, with an inter-trial interval of 15 min. In order to remove possible interfering scents, the mazes were cleaned with water and ethanol (70%) after each trial. 

#### 2.2.1. Open Field Test

The OF apparatus and methodology have been fully described elsewhere [13]. Briefly, at the beginning of the trial, each rat was placed in the center of the square arena (60 × 60 × 30 cm) and spontaneous exploration was recorded for 5 min. The following parameters were calculated: the cumulative duration in the center zone (CDCZ) (s), the frequency to the center zone (FCZ), the number of rearings, the total distance moved (TDM) (cm), and the percentage of time moving (%TM).

#### 2.2.2. Elevated Plus Maze Test 

The apparatus for the EPM test consisted of two open (50 × 20 cm) and two enclosed (50 × 20 × 30 cm) opposite arms elevated 100 cm above the floor. Each rat was placed in the center of the maze facing the open arm, and was allowed 5 min for free exploration. The parameters obtained in the EPM test were as follows: the cumulative duration in the open arms (CDOA) (s), the frequency to the open arms (FOA), the total distance moved (TDM) (cm), the percentage of time moving (%TM), the number of rearings, the number of head dippings, and the number of total exploratory activity (TEA) episodes. 

#### 2.2.3. Video Recording System and Analysis 

OF and EPM tests were recorded by using a digital video camera (SONY FDR AX33B) mounted above the mazes at a suitable height (150 and 250 cm, respectively). Interpretation of video files was conducted by Ethovision software XT 12 (Noldus Information Technology, the Netherlands). 

After completing behavioral testing, the rats were anaesthetized with a combination of ketamine (10 mg/kg, i.p.) and xylazine (5 mg/kg, i.p.), and sacrificed by decapitation. Brains were carefully removed and hippocampi were dissected. Afterwards, tissue samples were homogenized in phosphate buffered saline (PBS, 50 mM, pH 7.4) and frozen at −80 °C for further analysis. 

### 2.3. Oxidative Stress Markers Determination 

The homogenized samples of hippocampi were centrifuged (4000 rpm, 4 °C) for 15 min. The obtained supernatants were used to evaluate the following parameters of oxidative stress: index of lipid peroxidation, the enzymatic activities of superoxide dismutase (SOD) and catalase (CAT), and the level of total glutathione (GSH). 

The lipid peroxidation in the hippocampus was evaluated by following the method of Ohkawa and coworkers [14] and expressed as the level of thiobarbituric acid reactive substances (TBARS). The results were calculated using the standard curve of malondialdehyde (MDA) and presented as nanomoles of MDA per milligram of protein (nmol/mg protein). The activity of SOD was monitored by spectrophotometric measurement of inhibition of adrenalin decomposition to adrenochrome at 480 nm [15], while CAT activity was determined based on the rate of hydrogen peroxide decomposition at 240 nm [16]. The enzymatic activities were expressed as enzymatic units per milligram of protein (U/mg protein). The supernatants were also used for the determination of GSH levels in the hippocampus, using spectrophotometric assay based on the reaction with 5,5-dithio-bis-(2-nitrobenzoic acid) [17]. The results of GSH levels in tissue homogenates were expressed as milligrams of GSH per gram of protein (mg/g protein). Total protein concentrations were determined using the method of Lowry and coworkers [18] where bovine serum albumin was used as a standard. 

### 2.4. Quantification of Expression of Apoptotic Genes: RT PCR Analysis of Genes Involved in the Regulation of Cellular Apoptosis

Total RNA from the hippocampus was extracted using TRIzol reagent (Invitrogen, Waltham, MA, USA) according to the manufacturer’s instructions. For cDNA synthesis, we used 2 µg of total RNA with random hexamers and a High Capacity cDNA Reverse Transcription Kit (Applied Biosystems, Waltham, MA, USA). Real-time PCR was carried out using Thermo Scientific Luminaris Color HiGreen qPCR Master Mix (Applied Biosystems, Waltham, MA) and mRNA specific primers (Table 2) for *Bax*, *Bcl*-2, and β-*actin* as a housekeeping gene (Invitrogen, Waltham, MA). Real-time PCR reactions were done in the Mastercycler Ep Realplex (Eppendorf, Hamburg, Germany) and after data analysis, relative gene expression was calculated according to Livak and Schmittgen [19].

### 2.5. Statistical Analysis

Statistical analysis was performed with the SPSS version 20.0 statistical package (IBM SPSS Statistics 20, Chicago, IL, USA). The results are expressed as the means ± standard errors of the mean (SEM). Parameters were initially submitted to the Levene᾽s test for homogeneity of variance and to the Shapiro–Wilk test of normality. One-way ANOVA, followed by Bonferroni test was used for comparisons between the groups. Simple linear regression and Pearson᾽s coefficient of correlation were used to analyze relationships between parameters obtained in behavioral tests, oxidative stress markers, and relative gene expression. The significance was determined at *p* < 0.05 for all tests. 

## 3. Results 

As shown in Figure 1, cisplatin significantly lowered CDCZ and FCZ compared to the control group (F = 9.423 and 5.850, respectively, df = 3, *p* < 0.01). This anxiogenic feature of cisplatin was attenuated by simultaneous administration of NAC (*p* < 0.05, compared to the CIS group), which reversed CDCZ and FCZ almost to the control values. The exploratory activity in the OF test expressed by means of the number of rearings also confirmed the anxiogenic effect of cisplatin that was manifested by significant diminishing of vertical locomotion (F = 10.231, *p* < 0.01). NAC application along with cisplatin was sufficient to significantly increase that type of exploratory activity when compared to the CIS group (*p* < 0.05). Cisplatin administration also resulted in a significant decrease in locomotion parameters in the OF test (Figure 1D,E) of TDM and %TM (F = 8.552 and 12.551, respectively, *p* < 0.01) compared to the control values. Again, simultaneous administration of cisplatin and NAC diminished that anxiogenic impact of cisplatin on locomotor parameters (*p* < 0.05). When applied alone, NAC administration had no significant effect on the parameters from OF test when compared to the control. 

Like in the OF test, the anxiogenic response to cisplatin was manifested by a significant decline, compared to the control, in two principal indicators of anxiety in the EPM test: CDOA and FOA (F = 16.447 and 8.769, respectively, dF = 3, *p* < 0.01). As shown in Figure 2A,B, the beneficial effect of NAC on cisplatin-induced anxiogenic effects was expressed by a significant increase in CDOA (*p* < 0.01) and FOA (*p* < 0.05) in the combined group compared to CIS group. The parameters of locomotor activity in the EPM test (TDM and %TM; Figure 2C,D, respectively), were also significantly altered by the applied protocols (F = 12.827 and 6.335). The significant diminishing of locomotion was confirmed by the decline in both parameters in cisplatin treated rats when compared to the control (*p* < 0.01). The cisplatin-induced reduction of locomotion, similar to the results obtained in the OF test, was also abolished by simultaneous administration of NAC (*p* < 0.05), except for %TM. The exploratory activity in the EPM test, expressed as the number of rearings and head-dippings, as well as the number of TEA episodes (Figure 2E–G) was also significantly affected by the applied protocols (F = 18.475, 18.337 and 25.996, respectively). A single dose of cisplatin significantly reduced all three parameters of exploration compared to the control (*p* < 0.01). Although NAC supplementation was sufficient to attenuate the cisplatin-induced decline in those parameters compared to the CIS group (*p* < 0.05 and 0.01 for TEA), it was not able to restore the exploratory activity in EPM since it remained significantly above the control values (*p* < 0.05 and 0.01 for TEA). Similar to the OF test, the administration of NAC alone did not significantly affect any of parameters obtained in EPM test. 

The evaluation of oxidative stress parameters showed significant alteration of the TBARS level, as well as CAT and SOD activities (Figure 3A–C) in the hippocampus caused by the application of cisplatin and its combination with NAC (F = 40.872, F = 7.635, F = 10.372). Cisplatin administration induced a significant increase of the TBARS level (*p* < 0.01) and a significant decline in the examined antioxidant enzyme (CAT and SOD) activities (*p* < 0.01) in the hippocampus, while the GSH level in this group was almost unchanged (*p* > 0.05) compared to the control group. This prooxidative effect of cisplatin in the hippocampus was significantly attenuated by simultaneous NAC administration, resulting in a significant reduction (*p* < 0.01) of the TBARS level and increase of CAT (*p* < 0.05) and SOD (*p* < 0.05) activities, compared to the cisplatin-treated group. NAC application, when applied alone, had no significant effect on the studied oxidative stress markers in the hippocampus compared to the control group. As shown in Figure 3D, none of the applied protocols significantly altered total GSH levels (F = 1.757).

As shown in Figure 4A, none of the applied protocols had a significant impact on relative gene expression of proapoptotic *Bax* in rat hippocampi (F = 0.352, df = 3). However, the relative gene expression of hippocampal *Bcl*2 (Figure 4B) was significantly altered in this trial (F = 11.143). A single application of cisplatin significantly decreased the relative expression of this antiapoptotic gene compared to the control group (*p* < 0.01). This proapoptotic action of cisplatin was significantly attenuated (*p* < 0.05) by simultaneous NAC administration. This effect was also demonstrated by a significantly lower *Bax*/*Bcl*2 ratio (F = 12.866, *p* < 0.01) compared to cisplatin alone. Neither pro- nor anti-apoptotic relative gene expression in the rat hippocampus was significantly affected by NAC, compared to the control values, when applied alone. 

Simple regression analysis (Figure 5A–C) revealed that hippocampal lipid peroxidation, expressed as TBARS, significantly positively correlated with the relative expression of *Bax*, as well as the *Bax*/*Bcl*-2 ratio (Pearson᾽s r = 0.53 and 0.85, *p* = 0.002 and 5.9^−10^, respectively) but negatively (also significantly) with the relative expression of *Bcl*-2 in the rat hippocampus (r = 0.88, *p* = 4.4^−11^). In contrast, this analysis showed that the activity of both antioxidant enzymes, SOD (Figure 5D–F) and CAT (Figure 5G–I), significantly negatively correlated with *Bax* and the *Bax*/*Bcl*2ratio (r = 0.66 and 0.70, *p* = 3.3^−5^ and 9.3^−6^ for *Bax* and r = 0.74 and 0.73, *p* = 1.1^−6^ and 1.8^−6^ for the *Bax*/*Bcl*2 ratio, respectively). At the same time, the activity of both enzymes strongly (positively) correlated with the relative expression of *Bcl*2 (r = 0.86 and 0.68, *p* = 2.3^−10^ and 2.1^−5^, respectively). 

As shown in Figure 6, besides the impact on the relative hippocampal expression of pro- and anti-apoptotic genes, simple regression analysis revealed that the markers of oxidative stress also correlated with CDOA, the principle parameter of anxiety level obtained in the EPM test. The lipid peroxidation, expressed as TBARS, strongly and negatively correlated with the anxiolytic outcome in the EPM test (Figure 6A) as shown by the increase in CDOA (r = 0.82, *p* = 1.7^−8^). The opposite results were observed while testing the relationship between the activity of antioxidant enzymes, SOD and CAT, in the hippocampus, with CDOA (Figure 6B,C). SOD and CAT activity strongly and positively correlated with the indicator of decreased anxiety level expressed as CDOA (r = 0.73 and 0.70, *p* = 2.1^−6^ and 6.2^−6^ respectively).

The connectivity between the parameters that indicates apoptotic mechanisms and the behavioral outcome is also presented in Figure 7. The increase in the relative hippocampal expression of the (pro-apoptotic) *Bax* gene negatively correlated with CDOA (Figure 7A, r = 0.34, *p* > 0.05). Also, negative (but strong) correlation was observed for the *Bax*/*Bcl*-2 ratio and CDOA (Figure 7C, r = 0.71, *p* = 6.2^−6^). In contrast, the relative expression of the antiapoptotic gene *Bcl*-2 also strongly, but positively, correlated with the increase in CDOA, as an indicator of an anxiolytic-like effect (r = 0.84, *p* = 1.8^−9^). 

## 4. Discussion 

The potential beneficial effects of antioxidant supplementation are still very intriguing since a compromise must be made between the two principle key points of a platinum-based therapeutic approach: its ability to treat numerous malignancies and, at the same time, its numerous serious side effects that quite often limit therapeutic. A variety of antioxidant-containing sources (including vitamins, natural compounds, etc.) have been employed in investigations of cisplatin-induced toxicity prevention and/or cure, but there still has not been confirmation of the potential beneficial role of the drugs already approved for different clinical implications (that have been confirmed for their safety and wide dosage range), such as NAC, in the treatment of behavioral manifestations of cisplatin-induced neurotoxicity. 

Indeed, the results of this study showed that NAC, when applied on its own, did not affect any of the estimated parameters obtained in behavioral testing and hippocampus analysis for oxidative stress and apoptotic markers when compared to the values obtained in the control group. This is in line with a recent study [20], which presented the results of prolonged NAC administration (100 mg/kg for 20 consecutive days) that did not alter oxidative stress markers in the whole brain tissue samples by means of MDA, GSH, and total antioxidant capacity. Similar observations considering the impact on oxidative status in brain regions that were even more specific for the mood regulation (such as hippocampus) in rats were made in the extensive investigation of the other sulfur-containing amino acid, taurine, with no significant changes following continual SCAAs administration [21]. The lack of behavioral alterations after completing a 16 days trial with taurine (200 mg/kg), when compared to the control, was also reported in this study, and it is in accordance with the results of behavioral testing performed in our investigation. 

Unlike for the absence of significant alterations when NAC was applied alone, the administration of this SCAA performed in two doses (500 mg/kg, 5 days before and simultaneously with cisplatin application) resulted in an evident modulation of the estimated parameters induced by cisplatin. When applied alone, cisplatin produced clear a anxiogenic effect that was manifested by direct parameters of anxiety obtained in the OF, a decrease in CDCZ and FCZ (Figure 1A,B), and in EPM tests, a decrease in CDOA and FOA (Figure 2A,B). The anxiety-like behavior following cisplatin administration was confirmed by diminishing of exploratory activity in both the OF (Figure 1C) and EPM tests (Figure 2E–G). In addition, the anxiogenic response to cisplatin was also represented by means of decreased locomotor activity in both applied behavioral tests (Figure 1D,E and Figure 2C,D). The anxiety-like outcome of cisplatin administration observed in this study is in line with the results of our recent investigation [22], but the anxiogenic feature of cisplatin was also reported for single administration of cisplatin in a two-fold higher dose [21]. As expected, the cisplatin-induced anxiogenic effect was also described in a study with prolonged cisplatin administration (5 weeks) with a lower dose (5 mg/kg) compared to the dose applied in this study [23].

Beside the behavioral alterations, the treatment with cisplatin also significantly affected the hippocampal oxidative balance in this investigation. Namely, cisplatin administration resulted in an increase of lipid peroxidation (Figure 3A) and a decline in the antioxidant capacity, expressed by means of decreased activity of antioxidant enzymes, CAT (Figure 3B) and SOD (Figure 3C), with no significant alteration in the second-line defense system (GSH, Figure 3D). The results obtained in this study are in accordance with previously described increased MDA levels in the hippocampus, as well as with an observed diminishing of antioxidant enzymes activity [5,21]. However, the cisplatin protocol performed in this study failed to induce a significant decline of the GSH level as reported in those two studies. This obvious difference can be explained by either a lower dose (7.5 vs. 10 mg/kg) or shorter exposure (single dose vs. 7 weeks) to the chemotherapeutic in our investigation. 

The treatment with cisplatin, when applied alone, resulted in proapoptotic action in rat hippocampi in this study. Although the observed increase in *Bax* relative mRNA expression was not significant (Figure 4A), the diminishing of the antiapoptotic mechanism (Figure 4B) was sufficient to shift the pro-/antiapoptotic balance favoring increased apoptosis (Figure 4C). In general, our results are in line with the proapoptotic action of cisplatin previously reported in rodent hippocampi [8] and cell cultures [24], as well as in a human neuroblastoma cell line [25]. However, it should be noted that, unlike in the mentioned studies, the cisplatin-induced increase in proapoptotic markers was not significant. This difference could be attributed to a significantly lower dose of cisplatin applied in this investigation that was not sufficient to produce a more visible increase in proapoptotic markers levels in five days, suggesting that decline in antiapoptotic mechanisms may represent a much faster neuronal response to cisplatin. 

Based on the results obtained in this study, it is obvious that NAC supplementation along with cisplatin application (in the applied doses) was sufficient to diminish the action of cisplatin on the parameters of cisplatin-induced neurotoxicity estimated in this investigation. NAC administration attenuated the anxiogenic effect of cisplatin, resulting in the reversion of behavioral indicators obtained in both performed tests (Figure 1 and Figure 2) back to control values. This neuroprotective action of NAC was accompanied with the improvement of the hippocampal oxidative status (Figure 3), as well as with the amelioration of cisplatin-induced proapoptotic effects (Figure 4). The reduction of the anxiogenic response to cisplatin achieved by NAC supplementation, concomitant with enhanced antioxidative and antiapoptotic mechanisms, is in accordance with the previously described beneficial effects of antioxidant supplementation performed by various antioxidant sources, including SCAAs (taurine and d-Methionine) on cisplatin-induced neurotoxicity manifestations [21,26]. 

It seems that the antioxidative scavenger action of NAC, accompanied with the enhancement of the antioxidant defense capacity, may lead to the attenuation of the prooxidant action of cisplatin, which includes the cisplatin-induced alterations in gene expression, such as down regulation of Nrf2 (the principle protective factor against oxidative stress [27]) and consequent decrease in HO-1, and upregulation of NF-*κ*B. On this level, the oxidative stress additionally promotes deleterious effects of cisplatin by promoting proinflammatory cytokines release [28]. Once achieved, a neuroinflammatory cascade may lead to the previously described decline of hippocampal BDNF levels [29] with the consequent reduction of hippocampal volume that may contribute in cisplatin-induced behavioral alterations [30], such as the ones observed in this study. The postulated mechanism can be numerically supported by a strong correlation between lipid peroxidation and the activity of antioxidant enzymes in hippocampus with the most important behavioral indicator of anxiety observed in this investigation (Figure 6). According to the results obtained in this study, it seems that by preventing the oxidative damage, NAC may also reduce the cisplatin-induced DNA damage and therefore may attenuate the apoptotic action of cisplatin. As shown in Figure 5, the increased lipid peroxidation is accompanied by the decline of the antiapoptotic cell capacity in the hippocampus, while the decrease in antioxidative capacity strongly correlates with the prevalence of proapoptotic mechanisms. Since the estimated relationship between the pro-/antiapoptotic balance and the behavioral outcome strongly confirmed the anxiogenic response under the proapoptotic conditions (Figure 7), it is possible that this coherency is the basis for the connection between the apoptosis and the behavioral manifestations of neurological disorders [31]. 

Although there are various investigated brain regions responsible for mood regulation [32,33], the majority of studies in the field of behavioral alterations are focused on the hippocampus [34,35]. It is worth noticing that increased oxidative damage in all specific brain regions responsible for mood regulation correlates with mood disorders [36,37]. However, although the mentioned investigations match with our study on the basis of finally achieved mood alteration, it is hard to compare them with our results since the methodologies were completely different (mostly chronic, unpredictable mild stress) when compared to our iatrogenic model based on a chemotherapeutic agent. 

In the study of Zaki and coworkers [38] it is obvious that peripheral neurotoxicity was induced following prolonged cisplatin administration with a cumulative dose that was twice as high as the dose applied in this study (16 mg/kg vs. 7.5 mg/kg). A similar observation could be made for NAC supplementation. Namely, Zaki and coworkers applied NAC in a dose that was 50% higher than ours. According to those findings, it seems that a protocol with prolonged administration required higher doses of both cisplatin and NAC in order to produce significant alterations by means of peripheral neurotoxicity quantification. However, in line with this conclusion, a single administration of cisplatin and a shorter protocol with NAC were sufficient for the estimation of an acute response to cisplatin by means of the behavioral alterations. The complementarity of those two investigations was additionally confirmed on the basis of an increase in oxidative damage and apoptotic markers induced by cisplatin, which was diminished in both studies by applying antioxidant supplementation with NAC.

## 5. Conclusions 

According to the results obtained in this study, it seems that the antioxidant supplementation performed by NAC may attenuate the cisplatin-induced neurotoxicity manifestations. The underlying mechanism of the beneficial effects of NAC on the adverse cisplatin effects may include the decline in oxidative damage that down regulates increased apoptosis and reverses the anxiogenic action of cisplatin. Since the behavioral alterations may be considered as early signs of cisplatin-induced neurotoxicity, the NAC supplementation in this stage of therapy may be useful in the prevention of more profound neurological disorders that accompany the therapeutic protocols based on cisplatin. 

Our further investigations will address the estimation of different brain regions involved in the regulation of stress and anxiety (such as amygdala, paraventricular nucleus, etc.) to provide insights into the subtle mechanisms that can explain the neuroprotective role of sulfur-containing compounds, especially on the grounds of cisplatin-induced neurotoxicity. Furthermore, our future investigations in this field will cover the potential regional differences inside the targeted regions involved in mood regulation. Namely, according to our previous investigations, the possible mechanisms that connect behavioral alterations and the pathophysiological base of cisplatin action may be linked to very specific changes in mood controlling receptors, as well as to the amount of neurotrophic factors in a subregion of the hippocampus [39] instead of overall expression of mood regulators in each brain region responsible for mood regulation.

## Figures and Tables

**Figure 1 biomolecules-09-00892-f001:**
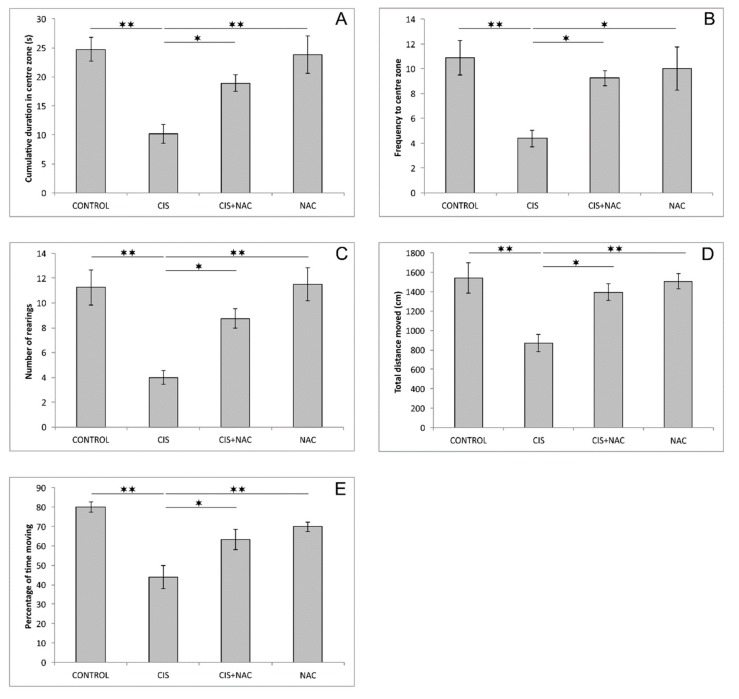
Parameters obtained in the open field test: (**A**) the cumulative duration in center zone, (**B**) the frequency to center zone, (**C**) the number of rearings, (**D**) the total distance moved, (**E**) the percentage of time moving. The values are mean ± standard error of the mean (SEM), * denotes a significant difference *p* < 0.05, ** denotes a significant difference *p* < 0.01.

**Figure 2 biomolecules-09-00892-f002:**
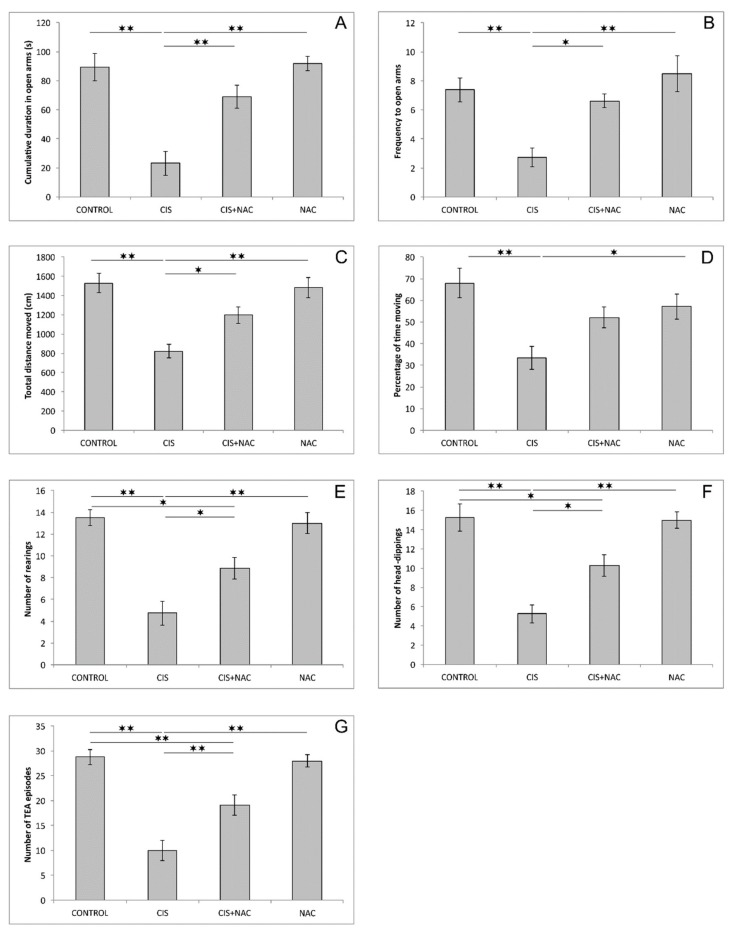
Parameters obtained in the elevated plus maze: (**A**) the cumulative duration in open arms, (**B**) the frequency to open arms, (**C**) the total distance moved, (**D**) the percentage of time moving, (**E**) the number of rearings, (**F**) the number of head-dippings, (**G**) the number of total exploratory activity (TEA) episodes. The values are mean ± standard error of the mean (SEM), * denotes a significant difference *p* < 0.05, ** denotes a significant difference *p* < 0.01.

**Figure 3 biomolecules-09-00892-f003:**
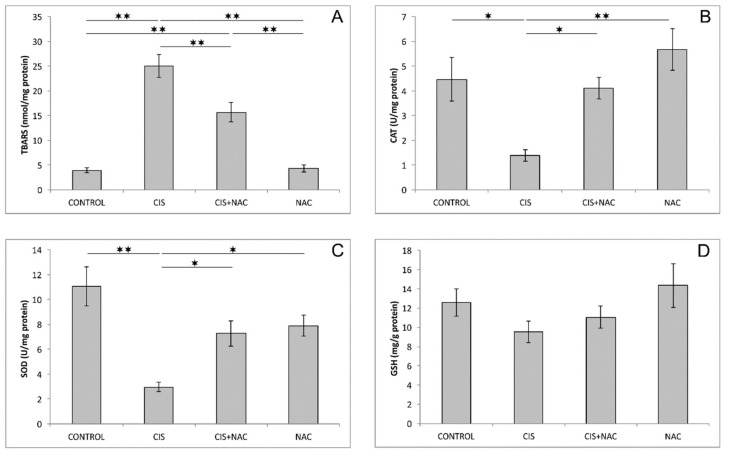
Oxidative stress parameters in rat hippocampus: (**A**) thiobarbituric acid reactive substances (TBARS), (**B**) catalase (CAT), (**C**) superoxide dismutase (SOD), (**D**) glutathione (GSH). The values are mean ± standard error of the mean (SEM), * denotes a significant difference *p* < 0.05, ** denotes a significant difference *p* < 0.01.

**Figure 4 biomolecules-09-00892-f004:**
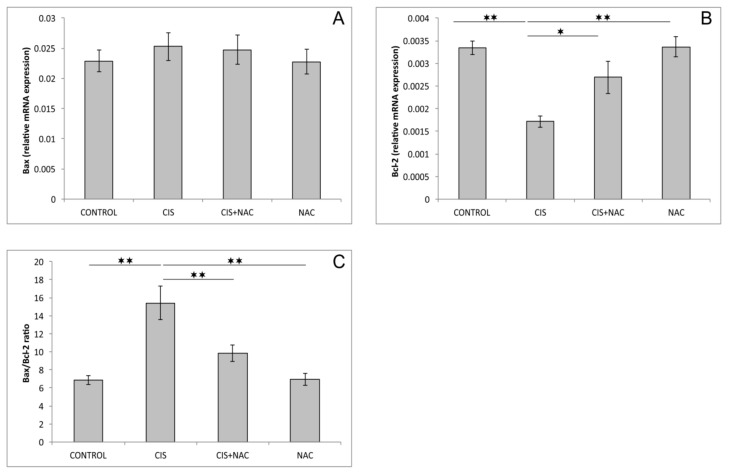
The relative gene expression of the pro- and anti-apoptotic genes in the rat hippocampus. (**A**) *Bax*, (**B**) *Bcl*-2, (**C**) *Bax*/*Bcl*-2 ratio. The values are mean ± standard error of the mean (SEM), * denotes a significant difference *p* < 0.05, ** denotes a significant difference *p* < 0.01.

**Figure 5 biomolecules-09-00892-f005:**
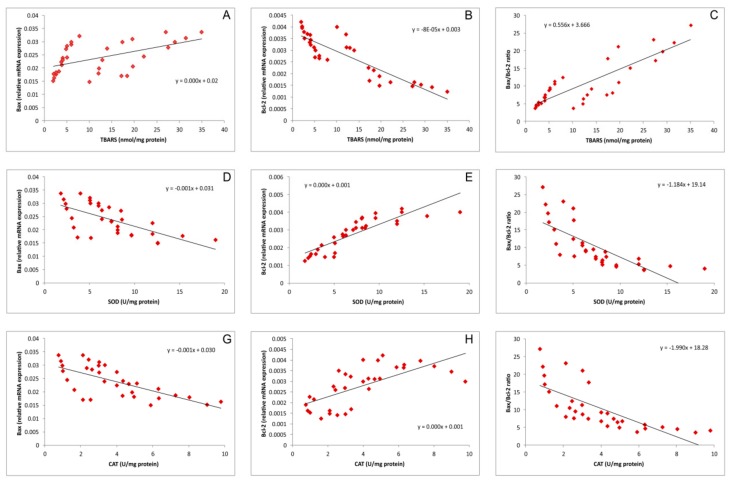
Relationship between the parameters of oxidative status in the rat hippocampus and the relative gene expression of apoptotic genes in the rat hippocampus for all investigated groups (**A**,**B**,**C**—represents the relationship between TBARS and *Bax*, *Bcl*-2 and *Bax*/*Bcl*-2 ratio, respectively; **D**,**E**,**F**—represents the relationship between SOD and *Bax*, *Bcl*-2 and *Bax*/*Bcl*-2 ratio, respectively; and **G**,**H**,**I**—represents the relationship between CAT and *Bax*, *Bcl*-2 and *Bax*/*Bcl*-2 ratio, respectively).

**Figure 6 biomolecules-09-00892-f006:**
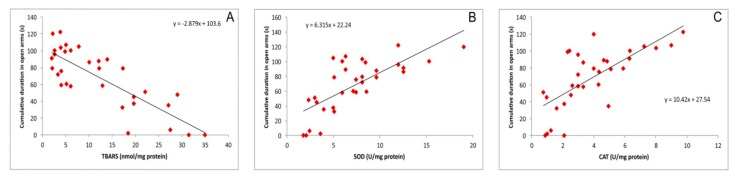
Relationship between the cumulative duration in the open arms and the hippocampal oxidative stress markers (**A**,**B**,**C**—represents the relationship between CDOA and TBARS, SOD, and CAT, respectively).

**Figure 7 biomolecules-09-00892-f007:**
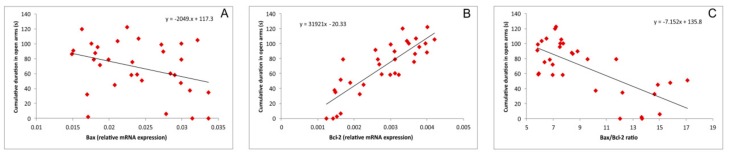
Relationship between the cumulative duration in the open arms and the relative expression of pro- and anti-apoptotic genes in the rat hippocampus (**A**,**B**,**C**—represents the relationship between CDOA and *Bax*, *Bcl*-2 and *Bax*/*Bcl*-2 ratio, respectively).

**Table 1 biomolecules-09-00892-t001:** Experimental protocol. CIS: cisplatin; NAC: *N*-acetylcysteine.

Group	Day 1	Day 5	Day 10
Control	saline (2 mL, i.p.)	saline (2 mL, i.p.)	testing and sacrifice
CIS	saline (2 mL, i.p.)	cisplatin (7.5 mg/kg, i.p.)	testing and sacrifice
NAC	NAC (500 mg/kg, i.p.)	NAC (500 mg/kg, i.p.)	testing and sacrifice
CIS + NAC	NAC (500 mg/kg, i.p.)	cisplatin (7.5 mg/kg, i.p.) + NAC (500 mg/kg, i.p.)	testing and sacrifice

**Table 2 biomolecules-09-00892-t002:** RT-PCR primers used in this study.

Name		Sequence (5′ to 3′)
β-*actin*	F	GATCAGCAAGCAGGAGTACGAT
	R	GTAACAGTCCGCCTAGAAGCAT
*Bax*	F	GCTACAGGGTTTCATCCAGGAT
	R	ATGTTGTTGTCCAGTTCATCGC
*Bcl*-2	F	GCAAAGCACATCCAATAAAAGCG
	R	GTACTTCATCACGATCTCCCGG

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
