# Peer review of "N-Acetylcysteine Protects against the Anxiogenic Response to Cisplatin in Rats"

_biomolecules, 2019, doi:10.3390/biom9120892_

Round 1

Reviewer 1 Report

The research subject studied by Vukovic et al. is not new as there is an excellent work of Zaki et al. about the protective effect of the NAC in on the
cisplatin-induced peripheral neurotoxicity (Folia Morphol.
Vol. 77, No. 2, pp. 234–245) . The Authors should compare their results with the once of Zaki et al. , especially please comment of the differences in the chosen doses.

Author Response

Response to reviewer 1

We would like to thank to the reviewer for positive comments about our work and we appreciate all suggestions that will improve the manuscript. Below we provide answers to their concerns and comply with their requests in the revision of the manuscript. All changes related to these requests are marked with track changes in the revised manuscript.

The research subject studied by Vukovic et al. is not new as there is an excellent work of Zaki et al. about the protective effect of the NAC in on the cisplatin-induced peripheral neurotoxicity (Folia Morphol. Vol. 77, No. 2, pp. 234–245). The Authors should compare their results with the once of Zaki et al., especially please comment of the differences in the chosen doses.

Although there are still ongoing research of peripheral neurotoxicity induced by chemotherapeutics (Zaki et al., 2018; Kanat et al., 2017), our previous investigations strongly confirmed that even following a single dose of cisplatin the behavioral manifestations of neurotoxicity were expressed even earlier, so we focused on that type of adverse effect induced by cisplatin.

We accepted the reviewer’s suggestion and performed the following changes in the manuscript:

In study of Zaki and coworkers [38] it is obvious that peripheral neurotoxicity was induced following prolonged cisplatin administration with cumulative dose that was twice as high when compared to the dose applied in this study (16 mg/kg vs. 7.5 mg/kg). A similar observation could be made for the NAC supplementation. Namely, Zaki and coworkers applied NAC in a dose that was 50% higher than ours. According to this, it seems that a protocol with prolonged administration required higher doses of both cisplatin and NAC in order to produce significant alterations by means of peripheral neurotoxicity quantification. However, in line with this conclusion, a single administration of cisplatin and shorter protocol with NAC were sufficient for the estimation of acute response to cisplatin by means of the behavioral alterations. The complementarity of those two investigations was additionally confirmed on the base of increase in oxidative damage and apoptotic markers induced by cisplatin, which was in both studies diminished by applying antioxidant supplementation with NAC.

Also, we tried to improve the English language and style.

Reviewer 2 Report

TITLE: N-acetylcysteine protects against the anxiogenic response to cisplatin in rats

The article focus in the anxiogenic effect of cisplatin (CIS) and their relationship with oxidative stress levels and pro-apoptotic effects in hippocampus. They tested the protective effects of N-acetylcysteine (NAC) administered in two doses one day before CIS treatment and other simultaneously. They used male Wistar albino rats randomized in 4 experimental groups and evaluated the levels of some oxidative stress markers in hippocampus: index of lipid peroxidation, superoxide dismutase (SOD) and catalase (CAT) activity and total levels of glutathione. In parallel, they checked by RT-PCR real time the expression of some apoptotic genes. The results show that CIS induces an increment in the oxidative stress markers and apoptotic indicators, that antioxidant NAC treatment reduces almost control levels. In addition, using behavioral test, they demonstrated that the anxiogenic behavior induces by CIS is also reverse with NAC treatment.

The manuscript is written correctly and the results are consistent but in my opinion, the approach and the results are far from being novel. The relationship between CIS treatment, anxiety  and high levels of oxidative stress markers is well known (Jangra, M. et al., European J of Pharmacol, vol. 791, pp. 51–61, 2016.).  In addition, the use of antioxidant compounds to reduces the deleterious effect of cisplatinun (including anxiety) is determinates before, including by authors (Kumburovic, I et al 2019. Oxid Med Cell Longev 8307196).  The authors must justify the novelty of their results in base on previous work.

In my opinion, they are an important aspect that will improve this paper. The authors used all hippocampus to check oxidative stress markers and apoptotic genes expression, and related this indicators with anxiety behavior,  but not all hippocampus is involve in anxiety. In fact, brain regions related with stress and anxiety include: central and medial amigdala, paraventricular nucleus of hypothalamus and ventral hippocampus (Wang C et al, 2019 Brain Res. Nov 15;1723:146392; Lovelock DF et al, 2019, Neuroscience.  Aug 25;418:50-58). In my opinion, this is not an accurate approximation. This work would improve if the levels of oxidative stress markers and apoptotic indicators were measured in amygdala, region highly involved in anxiety.

Minor comments

1.- The authors used the term hippocampal tissue, this is incorrect the tissue is nervous and hippocampus is a part of the brain. The correct term is hippocampus.

2.- The authors should justify the doses used of both CIS and NAC with any previous papers or dose response experiments.

3.- The last paragraph of 2.4 would be better in 2.1

Author Response

Response to reviewer 2

We would like to thank to the reviewer for positive comments about our work and we appreciate all suggestions that will improve the manuscript. Below we provide answers to their concerns and comply with their requests in the revision of the manuscript. All changes related to these requests are marked with track changes in the revised manuscript.

The article focus in the anxiogenic effect of cisplatin (CIS) and their relationship with oxidative stress levels and pro-apoptotic effects in hippocampus. They tested the protective effects of N-acetylcysteine (NAC) administered in two doses one day before CIS treatment and other simultaneously. They used male Wistar albino rats randomized in 4 experimental groups and evaluated the levels of some oxidative stress markers in hippocampus: index of lipid peroxidation, superoxide dismutase (SOD) and catalase (CAT) activity and total levels of glutathione. In parallel, they checked by RT-PCR real time the expression of some apoptotic genes. The results show that CIS induces an increment in the oxidative stress markers and apoptotic indicators, that antioxidant NAC treatment reduces almost control levels. In addition, using behavioral test, they demonstrated that the anxiogenic behavior induces by CIS is also reverse with NAC treatment.

The manuscript is written correctly and the results are consistent but in my opinion, the approach and the results are far from being novel. The relationship between CIS treatment, anxiety and high levels of oxidative stress markers is well known (Jangra, M. et al., European J of Pharmacol, vol. 791, pp. 51–61, 2016.).  In addition, the use of antioxidant compounds to reduces the deleterious effect of cisplatinun (including anxiety) is determinates before, including by authors (Kumburovic, I et al., 2019. Oxid Med Cell Longev 8307196). The authors must justify the novelty of their results in base on previous work.

To our knowledge, there are no literature data concerning the effects of sulfur-containing amino acids between chemotherapy, mood disorders, oxidative stress and apoptotic markers. In this study we consolidated all data related to NAC administration (alone and along with cisplatin), and tried to confirm possible mechanisms of action and to prove its neuroprotective role, beside all well known beneficial effects of that compound. 

As the reviewer mentioned, we previously published results concerning the effects of Satureja hortensis L. on cisplatin model of neurotoxicity established in our laboratory. Summer savory is a plant that has different bioactive constituents (but not sulfur-containing) and showed (beside all other) antioxidant properties. So, the aim of our work was to determine whether sulfur-based compounds (such as NAC) show some beneficial action in order to minimize a very common occurrence of neurotoxicity during treatment with chemotherapeutic agents (such as cisplatin).

We are fully aware that there were numerous attempts to minimize the cisplatin-induced oxidative damage in central nervous system by applying many of antioxidants that belong to various classes of antioxidants. The attenuation of cisplatin-induced oxidative damage was estimated in cisplatin-induced neurotoxicity using various vitamins such as thiamine pyrophosphate (Turan et al., 2014), riboflavin (Hassan et al., 2013), natural products with high content of antioxidants (walnut - Shabani et al., 2012; curcumin – Khadrawy et al.,2019; ginger - Mohd Sahardi et al., 2019), edaravone (Jangra et al., 2016), etc. However, the usage of sulfur-containing amino acids in the treatment of this kind of neurotoxicity may represent a new therapeutic approach. Additional confirmation for the lack of evidence that sulfur-containing amino acids may be beneficial in the treatment of behavioral manifestation of cisplatin-induced neurotoxicity could be found in our recent review article in Current Medicinal Chemistry (Rosic et al., 2018). An overview of sulfur-containing amino acids action on cisplatin-induced neurotoxicity was reported only with taurine, but with no behavioral estimation.

In my opinion, they are an important aspect that will improve this paper. The authors used all hippocampus to check oxidative stress markers and apoptotic genes expression, and related this indicators with anxiety behavior, but not all hippocampus is involve in anxiety. In fact, brain regions related with stress and anxiety include: central and medial amigdala, paraventricular nucleus of hypothalamus and ventral hippocampus (Wang C et al., 2019 Brain Res. Nov 15;1723:146392; Lovelock DF et al., 2019, Neuroscience.  Aug 25;418:50-58). In my opinion, this is not an accurate approximation. This work would improve if the levels of oxidative stress markers and apoptotic indicators were measured in amygdala, region highly involved in anxiety.

We accepted the reviewer’s suggestion and commented in Discussion section, as follows:

Although there are various investigated brain regions responsible for mood regulation [32, 33], the majority of studies in the field of behavioral alterations are focused on the hippocampus [34, 35]. It is worth noticing that increased oxidative damage in all specific brain regions responsible for mood regulation correlates with mood disorders [36, 37]. However, although the mentioned investigations match with our study on the base of finally achieved mood alteration, it is hard to compare it with our results since the methodology was completely different (mostly chronic unpredictable mild stress) when compared to our iatrogenic model based on chemotherapeutic agent.

Our further investigations will be addressed to the estimation of different brain regions involved in regulation of stress and anxiety (such as amygdala, paraventricular nucleus, etc.) in order to make full insight into subtle mechanisms that can explain the neuroprotective role of sulfur-containing compounds especially on the ground of cisplatin-induced neurotoxicity. Furthermore, our future investigations in this field will cover the potential regional differences inside the targeted regions involved in mood regulation. Namely, according to our previous investigations, the possible mechanisms that connect behavioral alterations and patophysiological base of cisplatin action may be linked to very specific changes in mood controlling receptors, as well as neurotrophic factors content in a subregion of hippocampus [33] instead of overall expression of mood regulators in each brain region responsible for mood regulation.

Minor comments

1.- The authors used the term hippocampal tissue, this is incorrect the tissue is nervous and hippocampus is a part of the brain. The correct term is hippocampus.

We accepted the reviewer’s suggestion and made changes in the manuscript.

2.- The authors should justify the doses used of both CIS and NAC with any previous papers or dose response experiments.

Our experimental results confirmed that the dose of cisplatin applied in the study is considered as non-toxic and safe (by recommendation of Prof. Wolfgang Dekant – Editor in Toxicology Letters). A single dose of cisplatin is commonly use for the induction of nephrotoxicity and dose range is between 15 mg/kg (Townsend et al., 2009) and 30 mg/kg (Mitazaki S et al., 2013). Also, dose of cisplatin for induction of hepatotoxicity can be even higher 40 mg/kg (Niu et al., 2017). Chronic protocols imply significantly higher doses of cisplatin.

It is well known that NAC is one of the most commonly used mucolytic drug, at the same time NAC has multiple protective mechanisms and could be considered as chemoprotective agent with special reference to lung cancer (van Zandwijk, 1995).  NAC as a chemoprotective agent showed antiapoptotic properties in small cell lung carcinoma cells (Wu et al., 2005). NAC applied in dose of 500 mg/kg did not show any kind of toxicity. Dose range for NAC administration is wide, and for example extreme higher doses are in use for treatment of addictive behavior (Tomko et al., 2018).

3.- The last paragraph of 2.4 would be better in 2.1

We agree with the reviewer and transferred the paragraph as suggested.

Also, we tried to improve the English language and style.

Round 2

Reviewer 2 Report

The manuscript has been significantly improved by authors.